# Tunable Color-Variable Solar Absorber Based on Phase Change Material Sb_2_Se_3_

**DOI:** 10.3390/nano12111903

**Published:** 2022-06-02

**Authors:** Xin Li, Mingyu Luo, Xinpeng Jiang, Shishang Luo, Junbo Yang

**Affiliations:** 1Center of Material Science, College of Liberal Arts and Sciences, National University of Defense Technology, Changsha 410073, China; lixin1010106@163.com (X.L.); jackson97666@163.com (X.J.); luoshishang_x@163.com (S.L.); 2Guangxi Key Laboratory of Multimedia Communications and Network Technology, School of Computer, Electronic and Information, Guangxi University, Nanning 530004, China; luomingyu20@163.com

**Keywords:** dynamic color-variable solar absorber, phase change material Sb_2_Se_3_

## Abstract

In this paper, a dynamic color-variable solar absorber is designed based on the phase change material Sb_2_Se_3_. High absorption is maintained under both amorphous Sb_2_Se_3_ (aSb_2_Se_3_) and crystalline Sb_2_Se_3_ (cSb_2_Se_3_). Before and after the phase transition leading to the peak change, the structure shows a clear color contrast. Due to peak displacement, the color change is also evident for different crystalline fractions during the phase transition. Different incident angles irradiate the structure, which can also cause the structure to show rich color variations. The structure is insensitive to the polarization angle because of the high symmetry. At the same time, different geometric parameters enable different color displays, so the structure can have good application prospects.

## 1. Introduction

As the most widely used and abundant clean energy, solar energy has always been the most popular among clean energy sources, and the utilization of solar energy has always been a hotspot for scientific research. Photovoltaic conversion [1,2,3,4] and photothermal conversion [5,6,7,8] are currently the main approaches for solar energy utilization. Currently, the conversion efficiency of photovoltaic systems is low, and the photoelectric conversion efficiency of one-dimensional structured solar cells is only around 30%. Solar cells with higher conversion efficiency can reach 47.1% by the combined action of multiple semiconductor materials with different band gaps [9], but they require near-perfect materials and multiple absorption band gaps. Solar photothermal conversion can be applied to thermophotovoltaic [10,11,12,13] and thermoelectric power generation [14,15,16], and catalytic hydrolysis [17,18,19,20]. At the same time, the utilization of light energy is higher, the material requirements are lower, and the equipment structure is simpler and has a low failure rate. A broadband solar absorber [21,22] is the key to solar thermal conversion, which directly affects the efficiency of solar thermal conversion systems and thermophotovoltaic systems. There are two main ways to improve the absorption capacity of solar absorbers: (1) increasing the absorption bandwidth, (2) increasing the absorption rate in the region of higher solar radiation capacity. At present, broadband absorbers still have problems, such as a narrow absorption bandwidth and single application. Therefore, it is necessary to design efficient and multifunctional broadband absorbers.

If only solar absorption is studied in the visible band, it would be rather limited. The addition of structural color studies can enrich the visible spectral studies. In general, the color does not change as long as the structure and dimensions of the microstructure and the optical properties of the constituent material do not change [23]. However, the color change of the conventional structural color requires changing the size of the structure to achieve the shift in the spectrum to cause the color change, which is a very challenging task. The emergence of phase change materials has changed this situation. Phase change materials have enormous potential for dynamic display [24] because they can change the internal atomic arrangement structure and thus switch between different phase states under optical, electrical, or thermal stimulation. During the phase transition, the physical properties of the material, such as force, heat, light, and electricity, are changed due to the change in the arrangement of atoms inside the material. Ge_2_Sb_2_Te_5_ (GST) has been widely studied as a representative of phase change materials [25,26]. The internal structure of GST is stable before and after the complete phase transformation. The phase transition of GST can change the optical properties of GST, and GST can be easily compounded with other materials. Recently, related studies have been proposed about GST nano-grating [27]. There are also related studies using ITO inserted into GST thin film layers to achieve a dynamic color display through ITO thickness modulation [28].

Recently, Sb_2_Se_3_ has attracted the attention of related researchers as a new phase change material due to its ultra-low loss characteristics [29,30]. Sb_2_Se_3_ was first studied as a thermoelectric semiconductor in the 1960s [29]. In recent years, it has been explored as an absorber material for solar cells due to its good photovoltaic property [31,32,33,34,35,36,37,38]. However, very little attention has been paid to Sb_2_Se_3_ as a phase change material. Matthew Delaney et al. studied the physical properties of Sb_2_Se_3_ phase change material and compared it with GST, and found that the refractive indices of the Sb_2_Se_3_ amorphous and crystalline states are closer to silicon, which is more favorable for integrated devices and better for achieving ultra-low loss in photonic integrated circuits [29]. The following year, Matthew Delaney et al. used Sb_2_Se_3_ integrated on a silicon-based surface to achieve low-loss programmable optical phase control in both the amorphous and crystalline states. This amply demonstrates the potential of Sb_2_Se_3_ as a phase change material in silicon-based integrated photonics [30]. Due to being a relatively new phase change material, there are almost no studies on the Sb_2_Se_3_ structure color. Therefore, relevant studies targeting the structural color of Sb_2_Se_3_ are very necessary.

In this paper, we propose the Ag/Sb_2_Se_3_/Al structure based on Sb_2_Se_3_ to achieve broadband absorption and dynamic color display in the visible. The transition between the amorphous and crystalline states brings dynamic color display to the structure. At the same time, the structure maintains high absorption under both the amorphous and crystalline states. The transition between amorphous and crystalline states is a process, and different crystalline components also present different color displays. For different incident angles, the structure can be made to show different color changes. Since the structure is highly symmetric, they are insensitive to the polarization angle. In addition, the different geometric parameters support different color displays. Therefore, the designed structure can achieve multifunctional compatibility in the visible band, i.e., it is compatible with solar absorption and color camouflage, and has great practical prospects.

## 2. Material and Structure

Figure 1 represents the structure designed based on the phase change material Sb_2_Se_3_. The structure from bottom to top is Al, phase change material Sb_2_Se_3_, Ag, and its structural parameters are *t*_1_ = 50 nm, *t*_2_ = 20 nm, *H* = 80 nm, length (*L*) = width (*W*) = 100 nm, period (*P*) = 300 nm. The underlying metal can act as a reflector so that the transmittance is 0. We use Lumerical software for finite-difference time-domain simulation. In the simulation, the *x*-axis and *y*-axis are set to periodic boundary conditions and the *z*-axis is set to perfect matching boundary conditions. Moreover, the light source is set as a plane wave light source with vertical incidence. This structure can be practically fabricated by conventional real-world processes such as magnetron sputtering and electron beam exposure. Structures of Ag and Al thin films deposited by magnetron sputtering are all very common. We can consider depositing an Al thin film by DC magnetron sputtering first, and then depositing a Sb_2_Se_3_ thin film by RF magnetron sputtering. Se is lost during the deposition of Sb_2_Se_3_ thin film, so co-sputtering of Sb_2_Se_3_ and Se is used in the sputtering process. Then, the Ag thin film is also deposited by DC magnetron sputtering, and the desired pattern needs to be etched by electron beam exposure technology. After the device is fabricated, the reversible transition between the crystalline and amorphous states of Sb_2_Se_3_ can be achieved by a low-energy laser pulse. The laser setting power is 90 mw, the pulse time from amorphous to crystalline is 100 ns, and the pulse time from crystalline to amorphous is 400 ns [29].

## 3. Results

Figure 2a shows the refractive index parameter of the Sb_2_Se_3_ amorphous and crystalline states in the visible band. We observe that the refractive index and extinction coefficient of the amorphous state are both smaller than those of the crystalline state. Therefore, the simulation calculation is carried out based on this refractive index parameter [29]. We use material parameters for metals Ag and Al from Palik [39]. Figure 2b shows the absorptivity and reflectivity spectrum in the visible (380 nm–780 nm). We find that the absorptivity of the amorphous state is higher than that of the crystalline state. The average absorptivity of the amorphous state is 85.43%, and the average absorptivity of the crystalline state is 75.25%.

To understand the optical properties of this structure, we explore its electromagnetic field distribution under the illumination of the light source at wavelength 380 nm–780 nm. As shown in Figure 3, we observe that the structure resonates at the same location regardless of the electromagnetic field under the amorphous and crystalline states. The magnetic field energy is mainly localized in the top Ag layer and the electric field energy is mainly localized around the top Ag layer. This is due to the incident light irradiating the surfaces of the structures Ag and Sb_2_Se_3_, and the free electrons absorbing energy, causing plasmon resonance and thus forming a strong electromagnetic field. We note that the electromagnetic intensity of the amorphous and crystalline states is almost the same. However, by calculating the average electric field under the amorphous and crystalline states, we find that the average electric field under the amorphous state, namely 0.63, is slightly higher than the average electric field under the crystalline state, which is 0.61. This also confirms that the absorption under the amorphous state is slightly higher than the absorption under the crystalline state, as shown in Figure 2. We also note that the refractive index under the crystalline state is higher than that under the amorphous state, but the average absorption under the crystalline state is smaller than that under the amorphous state, which is a phenomenon caused by the absorption edge effect [36].

In order to show that the structure absorbs sunlight energy, Figure 4a shows the sunlight energy absorption spectrum under standard AM1.5, for aSb_2_Se_3_ and cSb_2_Se_3_ states. We have observed that both aSb_2_Se_3_ and cSb_2_Se_3_ states can absorb most of the solar energy. Since the transition from the amorphous state to the crystalline state brings about a change in wavelength shift, this also causes a change in color display. Figure 4b represents the coordinates of the structure on the chromaticity diagram under the amorphous and crystalline states. Under the amorphous state, the structure appears blue, and the chromaticity coordinate is (0.2371, 0.3129). Under the crystalline state, the structure appears pink, and the chromaticity coordinate is (0.3374, 0.3645). Through the transformation from the amorphous state to the crystalline state, the structure can realize a transition from a cool tone to a warm tone.

The phase transition from aSb_2_Se_3_ to cSb_2_Se_3_ is a dynamic change process, which means that Sb_2_Se_3_ has different crystalline compositions. To further show the physical properties of Sb_2_Se_3_, we investigated the states with different Sb_2_Se_3_ crystalline compositions. The material parameters of Sb_2_Se_3_ with different crystalline fractions can be calculated by Equation (1) [40,41,42,43], where *ε_eff_*(*λ*) represents the dielectric constants of Sb_2_Se_3_ with different crystalline fractions, and *ε_c_*(*λ*) and *ε_a_*(*λ*) represent the dielectric constants of Sb_2_Se_3_ under the crystalline and amorphous states.
(1)$$\frac{\varepsilon_{eff}\left(\lambda \right)-1}{\varepsilon_{eff}\left(\lambda \right)+2}=m\times \frac{\varepsilon_{c}\left(\lambda \right)-1}{\varepsilon_{c}\left(\lambda \right)+2}+\left(1-m\right)\times \frac{\varepsilon_{a}\left(\lambda \right)-1}{\varepsilon_{a}\left(\lambda \right)+2}$$

As the crystallization fraction increases, we find that the two peak wavelengths are shifted separately. Figure 5b shows the coordinate positions of Sb_2_Se_3_ with different crystalline fractions on the chromaticity diagram. The crystallization fraction ranges from 0% to 100%, and the corresponding chromaticity coordinates are (0.2371, 0.3129), (0.2588, 0.338), (0.2926, 0.3507), (0.319, 0.3588), (0.3374, 0.3645). Through Figure 5c, we can observe the color display of different crystalline fractions very intuitively, which also makes the color change more dynamic and flexible.

To explore the adaptability of the structure, we change the angle of incident light based on the thin-film interference principle to achieve rich color responses. Figure 6a shows the reflectivity spectra and corresponding colors of different incidence angles under the aSb_2_Se_3_ state. As the incident angle changes, the trough is red-shifted and the peak is blue-shifted. Through calculation, we find that as the incident angle increases, the absorptivity first increases and then decreases. When the incident angle is lower than 50°, the absorptivity increases as the incident angle increases. When the incident angle is higher than 50°, the absorptivity decreases as the incident angle increases. Figure 6b represents the color coordinates corresponding to different incidence angles on the chromaticity diagram under the aSb_2_Se_3_ state. As the incident angle increases, the corresponding chromaticity diagram coordinates are (0.2371, 0.3129), (0.244, 0.3227), (0.2651, 0.3509), (0.2987, 0.3924), (0.3373, 0.4336), (0.3683, 0.4581), (0.3833, 0.4594), (0.3825, 0.443). We observe that as the incident angle increases, the color changes from a cool tone to a warm tone. Figure 7a shows the reflectivity spectra with different incident angles under the crystalline state. As the incident angle changes, the trough and peak are red-shifted. By calculation, as the incident angle increases, the absorptivity under the crystalline state changes in the same manner as under the amorphous state. Figure 7b represents the color coordinates corresponding to different incidence angles on the chromaticity diagram under the cSb_2_Se_3_ state. As the incident angle increases, the corresponding chromaticity diagram coordinates are (0.3374, 0.3645), (0.3405, 0.3685), (0.3497, 0.3801), (0.3642, 0.3985), (0.3825, 0.4211), (0.401, 0.4433), (0.4149, 0.459), (0.419, 0.4626). We find that as the angle of incidence increases, the warmer colors gradually deepen. Figure 8 shows the polarization spectra under the amorphous and crystalline states. The maximum and minimum values correspond to the extreme values of reflectivity, respectively. Due to the perfect symmetry of the structure, the designed structure is insensitive to any polarization of the incident light wave, which is crucial in practical applications.

The geometrical parameters of the structure are critical to the performance. Different functions can be realized by changing the geometric parameters. Therefore, we conduct further research on the geometric parameters. Figure 9a,c show the absorptivity spectra of different height (*H*) parameters under the amorphous and the crystalline states, respectively. We notice that under different height (*H*) parameters, the absorptivity changes significantly. Figure 9b,d show the structure colors of different height (*H*) parameters under the amorphous and crystalline states, respectively. Under the amorphous state, the height (*H*) is from 30 nm to 130 nm, and the chromaticity coordinates are (0.2373, 0.2622), (0.253, 0.3207), (0.2371, 0.3129), (0.1756, 0.10), (0.1862, 0.1433). Under the amorphous state, we observe that the color changes of the structure are all cool tones. Under the crystalline state, the height (*H*) ranges from 30 nm to 130 nm, and the chromaticity coordinates are (0.3078, 0.3257), (0.3621, 0.4086), (0.3374, 0.3645), (0.3035, 0.307), (0.2685, 0.2749), respectively. Under the crystalline state, we observe that the color change of the structure transitions between cool and warm tones. We have observed that the color changes are very obvious, regardless of whether the material is in the amorphous or crystalline state.

As shown in Figure 10a,c, we examine the absorptivity spectra of different length (*L*) parameters under amorphous and crystalline states. By calculating the absorption rate, we find that changing the length parameter has little effect on the long-wave absorption rate, while the short-wave absorption rate changes more significantly whether under the amorphous or crystalline state. As shown in Figure 10b, under the amorphous state, the colors displayed by the structure are all cool colors. Under the amorphous state, the length (*L*) is from 80 nm to 140 nm, and the chromaticity coordinates are (0.2125, 0.2373), (0.2182, 0.2606), (0.2371, 0.3129), (0.2473, 0.2946), (0.2259, 0.1635). As shown in Figure 10d, under the crystalline state, the colors displayed by the structure are all warm colors. Under the crystalline state, the length (*L*) is from 80 nm to 140 nm, and the chromaticity coordinates are (0.3081, 0.327), (0.3184, 0.3398), (0.3374, 0.3645), (0.3559, 0.3716), (0.352, 0.3261). In general, different geometric parameters can achieve rich colors.

## 4. Conclusions

In this paper, we design a solar absorber with variable color based on the phase change material Sb_2_Se_3_. In the visible band, the structure is able to absorb sunlight effectively. The average absorption under the amorphous state is slightly higher than that under the crystalline state through the combined effect of multiple resonance modes. The amorphous Sb_2_Se_3_ structure exhibits a cool color and the crystalline Sb_2_Se_3_ structure exhibits a warm color. The color changes significantly during the phase transition from the amorphous to the crystalline state. Therefore, the absorptivity can be judged by the color. Adjusting the geometrical parameters of the structure enables rich color variation to be achieved. The incident angle and polarization angle have positive feedback on the absorptivity and allow for rich color variation. Therefore, the structure has good prospects for application.

## Figures and Tables

**Figure 1 nanomaterials-12-01903-f001:**
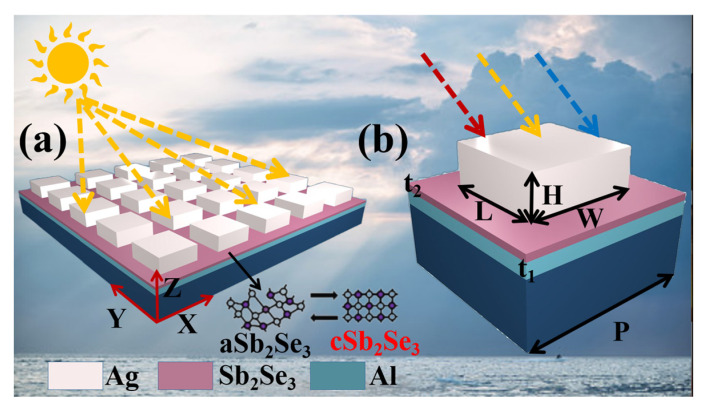
(**a**) Schematic diagram of the Sb_2_Se_3_-based metamaterial absorber; (**b**) unit periodic structure parameters.

**Figure 2 nanomaterials-12-01903-f002:**
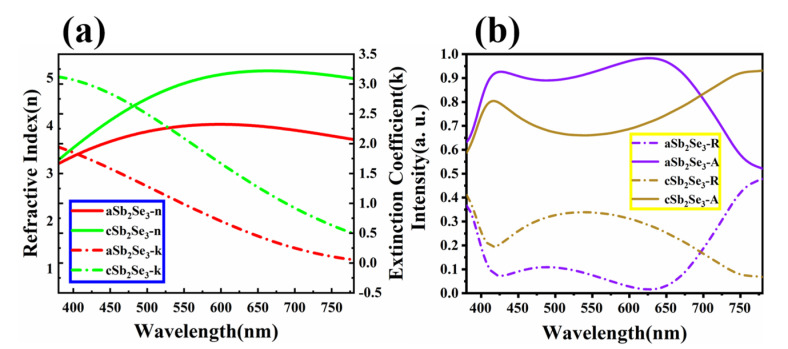
(**a**) Refractive index parameters of Sb_2_Se_3_ under amorphous and crystalline states; (**b**) absorptivity and reflectivity of the structure under amorphous and crystalline states.

**Figure 3 nanomaterials-12-01903-f003:**
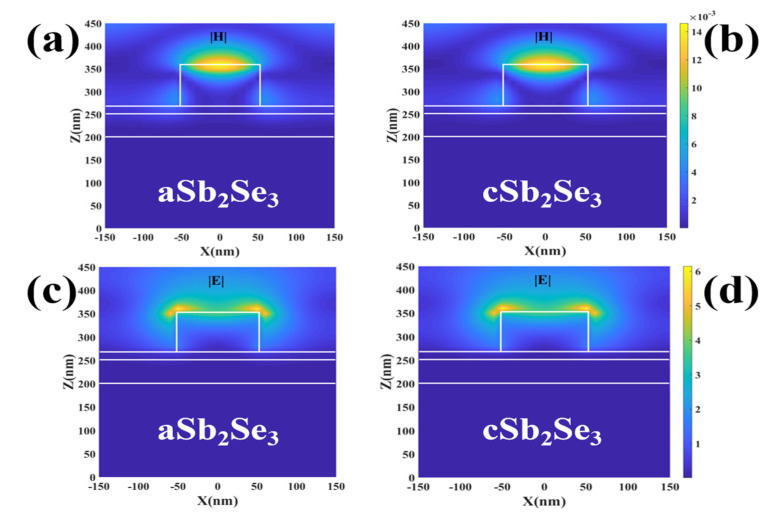
(**a**,**b**) Magnetic field distribution diagrams under the amorphous and crystalline states; (**c**,**d**) electric field distribution under the amorphous and crystalline states.

**Figure 4 nanomaterials-12-01903-f004:**
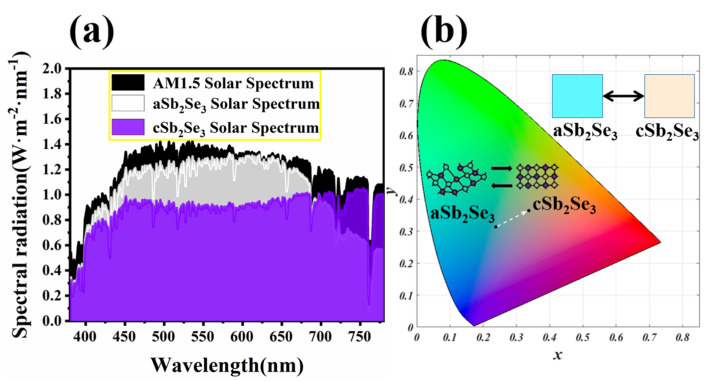
(**a**) Solar spectra under AM1.5, amorphous and crystalline states; (**b**) coordinates on the chromaticity diagram of the aSb_2_Se_3_ and cSb_2_Se_3_ states with their corresponding displayed colors.

**Figure 5 nanomaterials-12-01903-f005:**
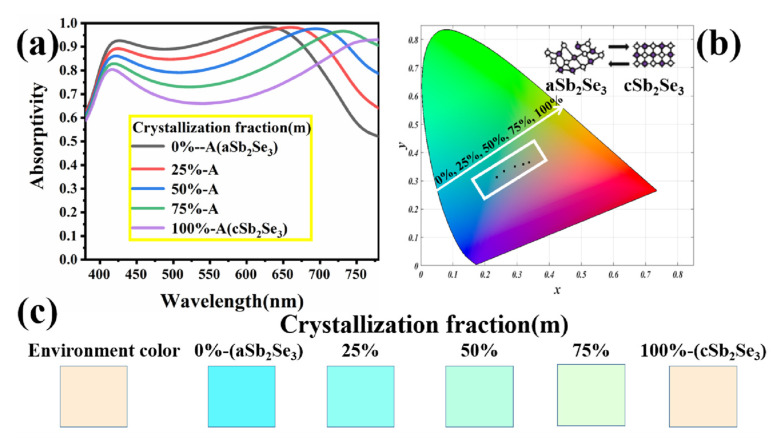
(**a**) Absorption spectra of different Sb_2_Se_3_ crystalline fractions; (**b**) coordinates of different Sb_2_Se_3_ crystalline fractions on the chromaticity diagram; (**c**) colors corresponding to different Sb_2_Se_3_ crystalline fractions.

**Figure 6 nanomaterials-12-01903-f006:**
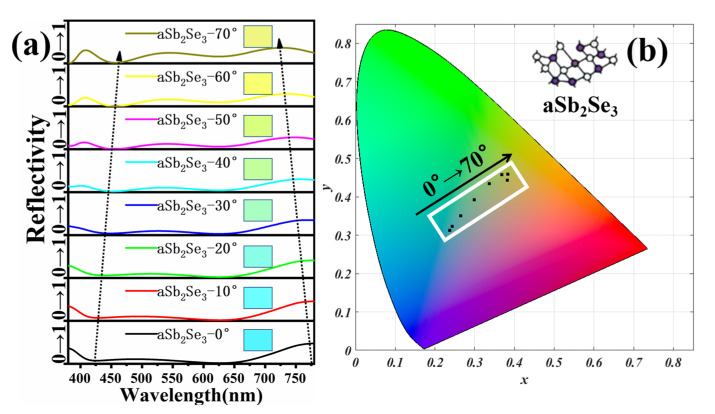
(**a**) Reflectivity spectra and corresponding colors of different incidence angles under aSb_2_Se_3_ state; (**b**) color coordinates corresponding to different incidence angles on the chromaticity diagram under aSb_2_Se_3_ state.

**Figure 7 nanomaterials-12-01903-f007:**
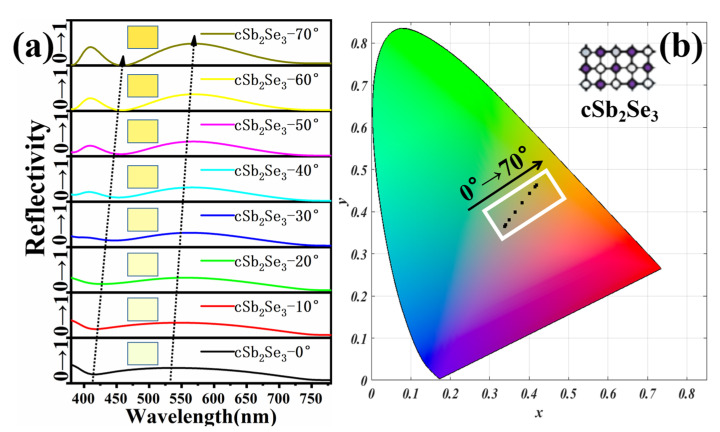
(**a**) Reflectivity spectra and corresponding colors of different incidence angles under cSb_2_Se_3_ state; (**b**) color coordinates corresponding to different incidence angles on the chromaticity diagram under cSb_2_Se_3_ state.

**Figure 8 nanomaterials-12-01903-f008:**
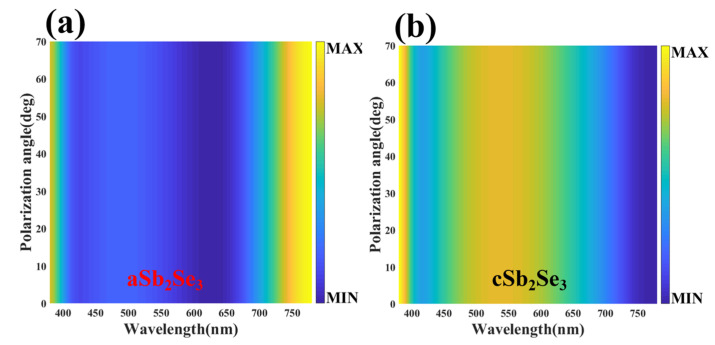
Polarization spectra under aSb_2_Se_3_ (**a**) and cSb_2_Se_3_ (**b**) states.

**Figure 9 nanomaterials-12-01903-f009:**
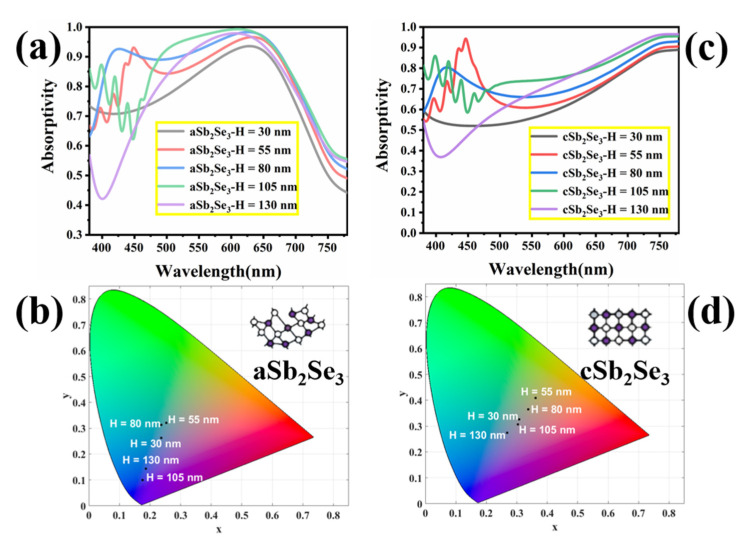
Absorption spectra (**a**) and corresponding chromaticity diagram coordinates (**b**) for different structural parameters (*H*) under the amorphous state; absorption spectra (**c**) and corresponding chromaticity diagram coordinates (**d**) for different structural parameters (*H*) under the crystalline state.

**Figure 10 nanomaterials-12-01903-f010:**
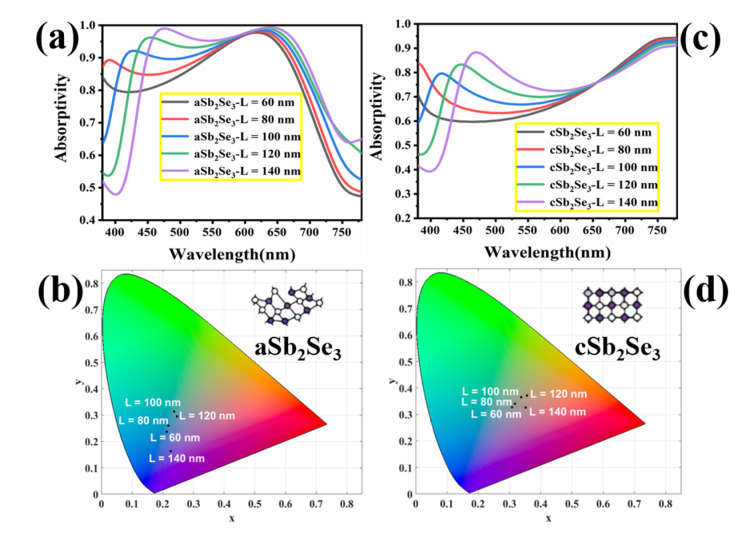
Absorption spectra (**a**) and corresponding chromaticity diagram coordinates (**b**) for different structural parameters (*L*) under the amorphous state; absorption spectra (**c**) and corresponding chromaticity diagram coordinates (**d**) for different structural parameters (*L*) under the crystalline state.

## Data Availability

The data presented in this study are available on request from the corresponding author.

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
