# Peer review of "Tunable Color-Variable Solar Absorber Based on Phase Change Material Sb_2_Se_3"

_nanomaterials, 2022, doi:10.3390/nano12111903_

Round 1
Author Response
Dear editor and reviewers:
We are grateful to the editor and reviewers for their constructive comments and suggestions on the revision of the manuscript. We have made all the necessary changes as suggested by the editor and reviewers. All the revisions in the manuscript and the supporting information have been highlighted in underline.
Reviewer #1:
Comment 1: lines 33-36 needs to be supported with references.
Respond: Thank you very much for your comments, we have added relevant references to the article.
Broadband solar absorber[21,22] is the key to solar thermal conversion, which directly affects the efficiency of solar thermal conversion systems and thermophotovoltaic systems.
References:
21.Lei, L.; Li, S.; Huang, H.; Tao, K.; Xu, P. Ultra-broadband absorber from visible to near-infrared using plasmonic metamaterial. Opt. Express. 2018, 26, 5686-5693.
22.Cai, Y.; Xu, K. D.; Feng, N.; Guo, R.; Lin, H.; Zhu, J. Anisotropic infrared plasmonic broadband absorber based on graphene-black phosphorus multilayers. Opt. Express. 2019, 27, 3101-3112.
Comment 2:lines 55-57 needs to be supported with references.
Respond: Thank you very much for your comments, we have added relevant references to the article.
Ge2Sb2Te5 (GST) has been widely studied as a representative of phase change materials[25, 26]. The internal structure of GST is stable before and after the complete phase transformation.
References:
[25]Gerislioglu, B.; Bakan, G.; Ahuja, R.; Adam, J.; Mishra, Y. K.; Ahmadivand, A. The role of Ge2Sb2Te5 in enhancing the performance of functional plasmonic devices. Mater. Today. Phys. 2020, 12, 100178.
[26]Mocanu, F. C.; Konstantinou, K.; Lee, T. H.; Bernstein, N.; Deringer, V. L.; Csányi, G.; Elliott, S. R. Modeling the phase-change memory material, Ge2Sb2Te5, with a machine-learned interatomic potential. J. Phys. Chem. B. 2018, 122, 8998-9006.
Comment 3:Numerical calculations: the authors do not indicate whether the finite-difference is a commercial software or ‘home’ made.
Respond: Thank you very much for your suggestion, the finite-difference we use is commercial software.
We use Lumerical software for finite-difference time-domain simulation.
Comment 4:The fabrication part in the methodology needs to be described in more detail. I also suggest an illustration of the fabrication process.
Respond: Thank you very much for your suggestion, we have revised the relevant part of the article.
Structures of Ag, Al thin films deposited by magnetron sputtering are all very common. We can consider depositing Al thin film by DC magnetron sputtering first, and then depositing Sb2Se3 thin film by RF magnetron sputtering. Se is lost during the deposition of Sb2Se3 thin film, so co-sputtering of Sb2Se3 and Se is used in the sputtering process. Then, the Ag thin film is also deposited by DC magnetron sputtering, and the desired pattern needs to be etched by electron beam exposure technology. After the device is fabricated, the reversible transition between the crystalline and amorphous states of Sb2Se3 can be achieved by low-energy laser pulse.
Comment 5:lines 101-103: the authors should indicate from where they took the values for the refractive index.
Respond: Thank you very much for your suggestion, about the reference about refractive index, we put it on line 117.
Therefore, the simulation calculation is carried out based on this refractive index parameter [29].
References
- Delaney, M.; Zeimpekis, I.; Lawson, D.; Hewak, D. W.; Muskens, O. L. A new family of ultralow loss reversible phase‐change materials for photonic integrated circuits: Sb2S3and Sb2Se3. Func. Mater. 2020, 30, 2002447.
Comment 6:Figure 2b. what is the thickness of the active area. I suggest to name the y-axis ‘reflectivity’rather than intensity.
Respond: Thank you so much for your suggestion. We have kept the material parameter thickness constant throughout the study. The thickness parameter of the relevant material is in "Material and Structure".Since the curve in Figure 2b not only has a reflectance curve, but also an absorptivity curve, we name the coordinates "Intensity".
Comment 7:Throughout the manuscript the authors show spectra in the range of 400-750 nm. Can they extend this to 400-1100 nm which better reflect the solar spectrum?
Respond: Thank you very much for your suggestion. Since our research focus on structure color, we choose visible wavelengths (380nm-780nm) based on human eye recognition. Since this band is also the main band where solar energy is concentrated, we study solar radiation briefly.
Comment 8: Figure 3. What is the wavelength of excitation? Is this under broadband illumination. Need to state this clearly.
Respond: Thank you very much for your suggestion, we have revised it in the article.
To understand the optical properties of this structure, we explore its electromagnetic field distribution under the illumination of the light source wavelength 380nm-780nm.
Comment 9: Please also add to following relevant light trapping papers:
DOI: 10.1038/NMAT3921, doi/10.1073/pnas.1008296107, doi: 10.1038/nmat2629,
doi.org/10.1016/j.nanoen.2019.04.082, DOI: 10.1002/solr.202100721
Respond: Thank you very much for your suggestion, we cite all references in the article.
DOI: 10.1038/NMAT3921
Brongersma, M. L.; Cui, Y.; Fan, S. Light management for photovoltaics using high-index nanostructures. Nat. Mater. 2014, 13(5), 451-460.
doi/10.1073/pnas.1008296107
Yu, Z.; Raman, A.; Fan, S. Fundamental limit of nanophotonic light trapping in solar cells. P. Natl. A. Sci. 2010, 107, 17491–17496.
doi: 10.1038/nmat2629
Atwater, H. A.; Polman, A. Plasmonics for improved photovoltaic devices. Nat. Mater. 2010, 9, 205–213.
doi.org/10.1016/j.nanoen.2019.04.082
Marko, G.; Prajapati, A.; Shalev, G. Subwavelength nonimaging light concentrators for the harvesting of the solar radiation. Nano. Energy. 2019, 61, 275–283.
DOI: 10.1002/solr.202100721
Chauhan, A.; Prajapati, A.; Calaza, C.; Fonseca, H.; Sousa, P. C.; Llobet, J.; Shalev, G. Near‐Field Optical Excitations in Silicon Subwavelength Light Funnel Arrays for Broadband Absorption of the Solar Radiation. Sol. RRL. 2021, 5, 2100721.
Thank you for your attention and patience, and if you have any questions, please don't hesitate to contact me.
Yours sincerely,
Junbo Yang

Reviewer 2 Report
In this work, the authors describe the differences in observed structural coloring between films of amorphous Sb2Se3 and crystalline Sb2Se3. I feel that this paper needs substantial improvements before it is ready for publication.
- The use of the word “dynamic” here is a buzzword that should be omitted. The authors do not demonstrate the reversibility of their process. Under such a liberal use of the word, any morphological change in any material that occurs by any process could be described as dynamic.
- The authors claim that for difference incident angles, the structure can be made to show difference color changes; while true, this just a basic principle of all thin film interference. The optical path difference of incident light changes as a function of the incident angle and so a series of colors is observed.
- The authors claim that “since the structure is highly symmetric, they are insensitive to polarization angle” It is not clear what symmetry they are referring to, or what such symmetry is affecting. In Figure 8, it is not clear what the min and max are – is this meant to be reflection? Finally, what do the authors mean here by the polarization angle? This term is usually used to refer to the Brewster’s angle, which depends on the index of refractions at an interface. Usually these kind of simulations plot the incidence angle, and there is a comparison between s- and p- polarization.
- The experimental details are extremely sparse. What are the sputtering parameters? How is the thickness of the layers determined? What is the electron beam exposure and etching process parameters – was this used to fabricate the patterned Ag? What caused the phase change- was it the e-beam or something like hot plate annealing? What is the substrate that the materials are deposited on? What instruments and parameters were used to measure refractive index, absorptivity, reflectivity, absorbance under solar spectra, chromaticity, etc.
- Are the authors sure that the silver layer was unaffected by their crystallization process? Thermal dewetting is commonly done to silver thin films in this temperature range to produce nanoparticles. Here, SEM imaging before and after crystallization should be provided.
- The authors claim specific crystallization fractions were achieved, but do not give the synthesis parameters used to achieve the different fractions. I am guessing that crystalline fractions are calculated based on effective index of refraction, but where is the data for the index of refraction of these different crystal phase fractions? In addition, without quantitative data such as XRD to prove crystalline phase fractions, there is really no proof that 100% crystallization was achieved in the first place so I don’t think the authors can claim this.
- The authors describe the formation of “warm” versus “cool” colors for amorphous versus crystalline material, but this is more properly descried as the sequence of colors commonly observed in thin film interference as the geometry and effective index of refraction of the layers are modified. Changing the thicknesses of the layers would have the same effect. See DOI: 10.1111/jmi.12641 for a visual of the expected sequence for a simple situation.
- The reason phase change materials are interesting is that the phase change is reversible. While Sb2Se3 has been demonstrated to be a phase change material under optical diode light, here no reversibility is demonstrated. If the authors could demonstrate the reverse phase change and resulting color change, these results would be much more significant, but they should at least acknowledge in the paper that this is a prototype study for what could be achieved by phase change using a different process. A mention in the introduction of how the phase change can be physically induced and reversed in Sb2Se3.
Author Response
Dear editor and reviewers:
We are grateful to the editor and reviewers for their constructive comments and suggestions on the revision of the manuscript. We have made all the necessary changes as suggested by the editor and reviewers. All the revisions in the manuscript and the supporting information have been highlighted in underline.
Reviewer #2:
Comment 1: The use of the word “dynamic” here is a buzzword that should be omitted. The authors do not demonstrate the reversibility of their process. Under such a liberal use of the word, any morphological change in any material that occurs by any process could be described as dynamic.
Respond: Thank you very much for your suggestion, we have revised it in the article.
Tunable color variable solar absorber based on phase change material Sb2Se3
Comment 2: The authors claim that for difference incident angles, the structure can be made to show difference color changes; while true, this just a basic principle of all thin film interference. The optical path difference of incident light changes as a function of the incident angle and so a series of colors is observed.
Respond: Thank you very much for your suggestion, we have made relevant revisions in the article.
Line 126: To explore the adaptability of the structure, we change the angle of incident light based on the thin-film interference principle to achieve rich color responses.
Comment 3: The authors claim that “since the structure is highly symmetric, they are insensitive to polarization angle” It is not clear what symmetry they are referring to, or what such symmetry is affecting. In Figure 8, it is not clear what the min and max are – is this meant to be reflection? Finally, what do the authors mean here by the polarization angle? This term is usually used to refer to the Brewster’s angle, which depends on the index of refractions at an interface. Usually these kind of simulations plot the incidence angle, and there is a comparison between s- and p- polarization.
Respond: Thank you very much for your comments, we have added relevant revisions to the article. The symmetry in the article refers to the symmetry of the structure, and this symmetry affects whether the structure is sensitive to the polarization angle. The angle of polarization here is not the academic Brewster angle, but the angle of polarization on the Lumerical software is the angle between the electric field direction of the light source in the normal plane of the light source propagation direction and the normal plane coordinate system. The maximum and minimum values in Figure 8 are the reflectivity of the structure. We have illustrated in the article.
Line 208: The maximum and minimum values correspond to the extreme values of reflectivity, respectively.
Comment 4: The experimental details are extremely sparse. What are the sputtering parameters? How is the thickness of the layers determined? What is the electron beam exposure and etching process parameters – was this used to fabricate the patterned Ag? What caused the phase change- was it the e-beam or something like hot plate annealing? What is the substrate that the materials are deposited on? What instruments and parameters were used to measure refractive index, absorptivity, reflectivity, absorbance under solar spectra, chromaticity, etc.
Are the authors sure that the silver layer was unaffected by their crystallization process? Thermal dewetting is commonly done to silver thin films in this temperature range to produce nanoparticles. Here, SEM imaging before and after crystallization should be provided.
Respond: Thank you very much for your suggestion. Due to the limited experimental conditions, we only conduct simulation studies on material parameters. However, regarding the experimental details, we have consulted relevant researchers to make the following modifications to the article. The substrate for material deposition is SiO2, the refractive index of the structure can be measured by an ellipsometer, and the absorptivity and reflectance can be measured by spectrophotometer. The use of low-energy laser pulse to switch the phase-change material between crystalline and amorphous states does not affect the Ag film.
Structures of Ag, Al thin films deposited by magnetron sputtering are all very common. We can consider depositing Al thin film by DC magnetron sputtering first, and then depositing Sb2Se3 thin film by RF magnetron sputtering. Se is lost during the deposition of Sb2Se3 thin film, so co-sputtering of Sb2Se3 and Se is used in the sputtering process. Then, the Ag thin film is also deposited by DC magnetron sputtering, and the desired pattern needs to be etched by electron beam exposure technology. After the device is fabricated, the reversible transition between the crystalline and amorphous states of Sb2Se3 can be achieved by low-energy laser pulse.The laser setting power is 90mw, the pulse time from amorphous to crystalline is 100ns, and the pulse time from crystalline to amorphous is 400ns [29].
References
- Delaney, M.; Zeimpekis, I.; Lawson, D.; Hewak, D. W.; Muskens, O. L. A new family of ultralow loss reversible phase‐change materials for photonic integrated circuits: Sb2S3and Sb2Se3. Adv. Func. Mater. 2020, 30, 2002447.
Comment 5: The authors claim specific crystallization fractions were achieved, but do not give the synthesis parameters used to achieve the different fractions. I am guessing that crystalline fractions are calculated based on effective index of refraction, but where is the data for the index of refraction of these different crystal phase fractions? In addition, without quantitative data such as XRD to prove crystalline phase fractions, there is really no proof that 100% crystallization was achieved in the first place so I don’t think the authors can claim this.
Respond: Thank you very much for your suggestion. I am sorry that due to the limited experimental conditions, our research is only a simulation study of the structure based on material parameters and no relevant experimental study has been conducted yet. As for the study of different crystallization fractions, theoretically we have used Equation 1 to study the crystallization fraction of the phase change material, where the source and basis of the equation is referred to the literature [41-43]. For the subsequent work we will consider experimentally corroborating our theoretical calculations.
Line 167: The material parameters of Sb2Se3 with different crystalline fractions can be calculated by Eq. (1) [41-43]
Comment 6: The authors describe the formation of “warm” versus “cool” colors for amorphous versus crystalline material, but this is more properly descried as the sequence of colors commonly observed in thin film interference as the geometry and effective index of refraction of the layers are modified. Changing the thicknesses of the layers would have the same effect. See DOI: 10.1111/jmi.12641 for a visual of the expected sequence for a simple situation.
Respond: Thank you very much for your comments, we have added relevant revisions to the article.
Line 198:We observe that as the incident angle increases, the color changes from cool tone to warm tone.
Line 206:We find that as the angle of incidence increased, the warmer colors gradually deepened.
Line 228:Under the amorphous state, we observe that the color changes of the structure are all cool tones.
Line 231:Under the crystalline state we observe that the color change of the structure transitions between cool and warm tones.
Comment 7: The reason phase change materials are interesting is that the phase change is reversible. While Sb2Se3 has been demonstrated to be a phase change material under optical diode light, here no reversibility is demonstrated. If the authors could demonstrate the reverse phase change and resulting color change, these results would be much more significant, but they should at least acknowledge in the paper that this is a prototype study for what could be achieved by phase change using a different process. A mention in the introduction of how the phase change can be physically induced and reversed in Sb2Se3
Respond: Thank you very much for your suggestion. There has been a preliminary study of Sb2Se3 phase change materials in silicon-based optoelectronics. The reversibility of the Sb2Se3 phase change material has been verified in reference 29. It is mentioned in the article that Sb2Se3 was in the amorphous state when the sample preparation was completed. To verify the reversible conversion of crystalline and amorphous states, laser focusing was used in the article. The laser power is set to 90 mw, and the pulse time from amorphous to crystalline state is 100 ns, and the pulse time from crystalline to amorphous state is 400 ns. Therefore, based on the reversibility of this method, we use the material parameters given in the article for the related simulation study. The method of conversion between amorphous and crystalline states has been explained in detail in my article.
After the device is fabricated, the reversible transition between the crystalline and amorphous states of Sb2Se3 can be achieved by low-energy laser pulse. The laser setting power is 90mw, the pulse time from amorphous to crystalline is 100ns, and the pulse time from crystalline to amorphous is 400ns [29]
Thank you for your attention and patience, and if you have any questions, please don't hesitate to contact me.
Yours sincerely,
Junbo Yang

Round 2
Reviewer 2 Report
The paper is more clear now after the revisions made by the authors and I feel it is suitable for publication.
I would still recommend the authors a small but critical edit to their text, in section 2 "Material and Structure" Basically, instead of saying "The structure IS first deposited....", instead say "This structure CAN BE practically fabricated by conventional real-world processes such as ..". This makes absolutely clear to the reader that these processes are not being directly used in this study.
Author Response
Dear editor and reviewers:
We are grateful to the editor and reviewers for their constructive comments and suggestions on the revision of the manuscript. We have made all the necessary changes as suggested by the editor and reviewers. All the revisions in the manuscript and the supporting information have been highlighted in underline.
Reviewer #2:
Comment 1: I would still recommend the authors a small but critical edit to their text, in section 2 "Material and Structure" Basically, instead of saying "The structure is first deposited....", instead say "This structure can be practically fabricated by conventional real-world processes such as ..". This makes absolutely clear to the reader that these processes are not being directly used in this study.
Respond: Thank you very much for your suggestion, we have revised it in the article.
This structure can be practically fabricated by conventional real-world processes such as magnetron sputtering and electron beam exposure.
Thank you for your attention and patience, and if you have any questions, please don't hesitate to contact me.
Yours sincerely,
Junbo Yang
